materials science/chemical physics

GeSe, electronic structure, thermoelectric properties, lattice thermal conductivity, relaxation time

**Author for correspondence:**
Jianhui Yang
e-mail: yjh20021220@foxmail.com

# Thermoelectric performances for both p- and n-type GeSe

Qiang Fan[1], Jianhui Yang[2], Jin Cao[2] and Chunhai Liu[3]

[1]School of Electronic and Material Engineering, and [2]School of Mathematics and Physics, Leshan Normal University, Leshan 614004, Sichuan, People's Republic of China
[3]College of Materials and Chemistry and Chemical Engineering, Chengdu University of Technology, Chengdu 610059, People's Republic of China

 QF, 0000-0002-3059-2436; JY, 0000-0001-7875-4353

In this paper, the thermoelectric properties of p-type and n-type GeSe are studied systematically by using first principles and Boltzmann transport theory. The calculation includes electronic structure, electron relaxation time, lattice thermal conductivity and thermoelectric transport properties. The results show that GeSe is an indirect band gap semiconductor with band gap 1.34 eV. Though p-type GeSe has a high density of states near Fermi level, the electronic conductivity is relative low because there is no carrier transport pathway along the *a*-axis direction. For n-type GeSe, a charge density channel is formed near conduction band minimum, which improves the electrical conductivity of n-type GeSe along the *a*-axis direction. At 700 K, the optimal ZT value reaches 2.5 at $4 \times 10^{19}$ cm$^{-3}$ for n-type GeSe, while that is 0.6 at $1 \times 10^{20}$ cm$^{-3}$ for p-type GeSe. The results show n-type GeSe has better thermoelectric properties than p-type GeSe, indicating that n-type GeSe is a promising thermoelectric material in middle temperature.

## 1. Introduction

Thermoelectric material, as a functional material that can directly convert thermal energy to electrical energy without mechanical components, plays an important role in environmental pollution control and energy crisis resolution. The thermoelectric conversion efficiency of thermoelectric material can be described by the dimensionless thermoelectric figure of merit (ZT), ZT = $S^2\sigma T/(k_l + k_e)$, where $S$, $\sigma$, $k_l$, $k_e$ and $T$ represent Seebeck coefficient, electrical conductivity, lattice thermal conductivity, electronic thermal conductivity and absolute temperature, respectively. High power factor (PF = $S^2\sigma$), low thermal conductivity is beneficial to obtain high ZT value. Because the Seebeck coefficient, electrical conductivity and electronic thermal conductivity are related to the electronic structure and carrier concentration, it is difficult to adjust and control a certain parameter alone to improve the thermoelectric properties of the

material. In recent years, the efforts to enhance the ZT value mainly focus on improving the power factor through energy band engineering and reducing the thermal conductivity through nanometre or superlattice [1–6]. Both theoretical and experimental research shows that these methods can effectively improve ZT value, but it is difficult to be applied for the difficulty in preparation. The bulk material, with non-toxic elements, high stability, easy to prepare and reasonable thermoelectric properties has always been the unremitting target of researchers.

In 2014, Zhao et al. [7] reported that single-crystal SnSe has excellent thermoelectric performance for the first time, and the maximum ZT value at 923 K reaches 2.6 along the b-axis. The main reason for the excellent thermoelectric performance of single-crystal SnSe is the strong phonon anharmonicity, which leads to very low lattice thermal conductivity [8]. The excellent thermoelectric properties of SnSe promote the development of thermoelectric properties of group IV–VI chalcogenides at middle temperature. As a homologous compound of SnSe, GeSe has attracted much attention [9–18]. Hao et al. [19] calculated the GeSe electronic structure using VASP software with generalized gradient approximation (GGA) of Perdew–Burke–Ernzerhof (PBE) exchange functional, and comparative study of the thermoelectric performance of GeSe and SnSe with the Boltzmann transport theory. They reported the ZT value of p-type GeSe along the b-axis is even higher than that of SnSe at the same carrier concentration. The calculated band gap of GeSe is 0.85 eV, which is lower than the experimental value due to the underestimation of the band gap by GGA-PBE functional. Based on Boltzmann transport theory, Hao et al. obtained the ratio of electrical conductivity to relaxation time ($\sigma/\tau$) and Seebeck coefficient. In order to gain electronic conductivity, Hao et al. [19] determined the carrier concentration by comparing the calculated Seebeck coefficient of GeSe with the measured Seebeck coefficient of SnSe [20]. The relaxation time $\tau$ was determined by comparing the experimental measured $\sigma$ of SnSe with the calculated $\sigma/\tau$ of GeSe at the same carrier concentration. Experimental study shows that the ZT value of p-type GeSe is only 0.2 at 700 K due to the difficulty in obtaining optimal carrier concentration (only up to approx. $10^{18}$ cm$^{-3}$) [21]. The high doping concentration predicted in theory cannot be obtained in the experiment, which restricts the development of GeSe as a p-type thermoelectric material. Recently, Cui et al. [11] found that n-type GeSe has higher electrical conductivity than p-type GeSe in the a-axis direction (interlayer direction), and GeSe is also a potential n-type thermoelectric material. However, the relaxation time and thermal conductivity were not considered in that paper. Up to now, the research on the thermoelectric performance of GeSe is not complete. In this paper, we do a systematic study on the electronic structure, electron transport properties, lattice thermal conductivity of GeSe to strengthen the understanding of the thermoelectric performance of GeSe, and promote its application in the thermoelectric field.

# 2. Computational details

The thermodynamic stable structure of GeSe at atmospheric pressure and low temperature is an orthorhombic (Pnma, no. 62) structure. The phase transition from Pnma to Fm-3m occurs at 907 K [22], and the high-pressure phase transition from Pnma to Cmcm occurs at 37 GPa [23]. In this paper, the thermoelectric properties of GeSe are studied at atmospheric pressure and medium temperature. The crystal structure is shown in figure 1 with VESTA [24]. As can be seen from figure 1, there are eight atoms in the unit cell; the strong GeSe covalent bond is formed in the b–c plane; layer structure is formed in the a-axis direction, and the two layers are combined together by the weak van der Waals force.

The electronic structure is optimized by using the VASP software based on density functional theory [25]. The PBE functional under GGA approximation is selected to optimize the geometry structure, and calculate total energy. The van der Waals force is considered by optB88-vdW functional [26]. The cut-off energy of the plane wave is set to 600 eV. According to the convergence test, the Monkhorst–Pack method is used for k-point sampling in the first irreducible Brillouin zone, and $5 \times 15 \times 15$ k-point grid is selected. In geometric optimization and self-consistent calculations, the self-consistent precision is set as energy per atom converges to $1 \times 10^{-6}$ eV, and the maximum force per atom is less than 0.001 eV Å$^{-1}$. In order to overcome the underestimation of the band gap by GGA-optB88 and obtain more accurate electronic structure, the electronic structure is calculated by HSE06 functional [27].

Based on the energy eigenvalue calculated from the first principle, the electron transport properties are calculated by BoltzTraP software [28] based on the semi-classical Boltzmann transport theory. The electron relaxation time $\tau$ is treated as constant, and the rigid energy band model is used to simulate

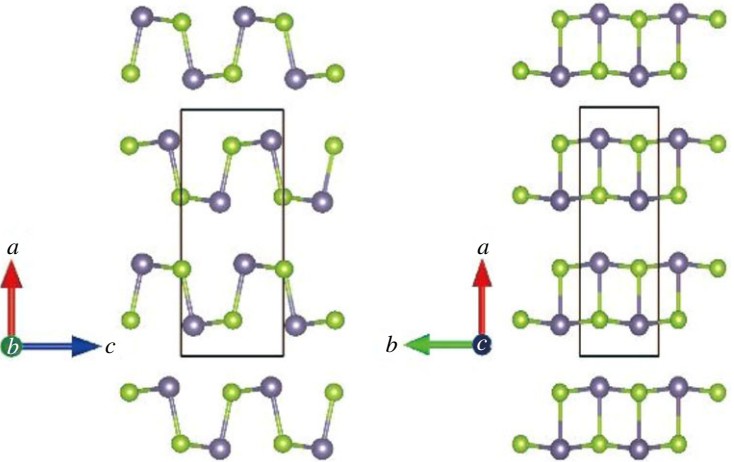

**Figure 1.** Crystal structure of GeSe (Pnma), grey represents Ge atom and green represents Se atom.

the doping concentration by changing the chemical potential position without considering the influence of doping on the band structure.

Since the Boltzmann transport theory can only calculate $\sigma/\tau$, but not $\sigma$, it is necessary to know the electron relaxation time $\tau$ in order to further calculate the ZT value. However, due to the complex scattering mechanism, the relaxation time $\tau$ is usually difficult to determine. In many theoretical calculations, the electron relaxation time is obtained by comparing the calculated $\sigma/\tau$ at a certain temperature and carrier concentration with the experimental data [29,30], or directly take $\tau$ as a constant of femtosecond magnitude [31,32]. In this paper, the deformation potential theory [33,34] is used to calculate the electron relaxation time $\tau$. In three-dimensional materials, the mobility $\mu_\beta$ and relaxation time $\tau$ along any $\beta$ direction are defined as

$$\left.\begin{aligned} \mu_\beta &= \frac{2\sqrt{2\pi}eC_\beta\hbar^4}{3\left(k_B T m_{\mathrm{dos}}^*\right)^{3/2} m_\beta^* E_{1\beta}^2} \\ \tau_\beta &= \frac{\mu_\beta m_\beta^*}{e}. \end{aligned}\right\} \tag{2.1}$$

and

where $C_\beta$, $E_{1\beta}$ are elastic constant and deformation potential along $\beta$ direction, respectively. $C_\beta$ is defined as the $C_\beta = 1/V_0\, \partial^2(E_t - E_0)/\partial(\Delta l/l_0)^2|_{l=l_0}$, which is obtained from total energy change $(E_t - E_0)$ corresponds to the change $\Delta l/l_0$ as a quadratic function fitting. $V_0$ is the undeformed unit cell volume. $E_{1\beta}$ is defined as the band edge variation under strain $(E_{1\beta} = \Delta E/(\Delta l/l_0))$. $m_{\mathrm{dos}}^*$ is the effective mass of state density, defined as $m_{\mathrm{dos}}^* = \sqrt[3]{m_a^* m_b^* m_c^*}$, where $m_a^*$, $m_b^*$, $m_c^*$ are effective mass along the $a$, $b$ and $c$ directions, respectively. The effective mass is obtained by $1/m^* = 1\partial^2 E/(\hbar^2\partial k^2)$, where $E$ is the energy eigenvalue of the valence band maximum (VBM) or the conduction band minimum (CBM). $\hbar$ and $k$ are reduced Planck constant and wavevector, respectively.

The calculation of lattice thermal conductivity is an issue which has received a lot of attention. The main calculation methods include molecular dynamics, third-order force constant and various semi-classical approximation. In this paper, we use semi-classical approximation methods to calculate lattice thermal conductivity based on phonon spectrum. The finite displacement method is used to calculate the phonon spectrum by using the PHONOPY software [35]. In the calculation of atomic force, $2 \times 3 \times 3$ supercell including 144 atoms is considered, and the atomic force is calculated using VASP software. In the calculation of atomic force, the k-point sampling in the first irreducible Brillouin zone of the supercell is selected as $1 \times 2 \times 2$ Monkhorst–Pack space sampling. The cut-off energy and self-consistent condition are consistent with the self-consistent calculation in the unit cell. According to the phonon dynamics theory, the lattice thermal conductivity can be expressed as follows:

$$k(T) = \frac{1}{3}\int C_V(\omega, T) v_g^2(\omega)\tau(\omega, T) g(\omega)\, \mathrm{d}\omega, \tag{2.2}$$

where $\omega$ is the phonon angular frequency; $C_V(\omega, T)$, $v_g(\omega)$, $\tau(\omega, T)$ and $g(\omega)$ denote the mode-specific heat, phonon group velocity, phonon relaxation time and phonon density of state distribution function at temperature $T$, respectively. In this work, we consider the contribution of all phonon branches to the

thermal conductivity, and a finite number of k-points are considered in the calculation of the thermal conductivity in the principal axis direction. The thermal conductivity tensor along the principal direction can be expressed as [36]

$$k_{ii}(T) = \sum_{\alpha=1}^{3N} \sum_{\beta=1}^{M_k} C_V(\omega_{ii}^{\alpha,\beta}, T) \left[ v_g\left(\omega_{ii}^{\alpha,\beta}\right) \right]^2 \tau\left(\omega_{ii}^{\alpha,\beta}, T\right) W\left(\omega_{ii}^{\alpha,\beta}\right). \tag{2.3}$$

The summation goes over all k-points ($M_k$) and all $3N$ phonon branches, where $N$ is the number of atoms in the unit cell, $W$ is the weight of k-point, and the total weight is normalized to 1,

$$\sum_{\beta=1}^{M_k} W(\omega_{ii}^{\alpha,\beta}) = 1. \tag{2.4}$$

In this work, we use finite number of k-points in the first irreducible Brillouin zone, i.e. 201 k-points are selected in [100], [010] and [001] directions.

Under the relaxation time approximation, the different scattering mechanisms are decoupled from each other. The total relaxation time $\tau(\omega, T)$ and the relaxation time of various scattering are given by [37]

$$\tau^{-1}(\omega,T) = \sum_i \tau_i^{-1}(\omega, T). \tag{2.5}$$

We consider Umklapp scattering and boundary scattering in our calculation, and which are given by

$$\left.\begin{aligned} \tau_U(\omega,T) &= \lambda \frac{M}{a} \frac{v_g(\omega)v_p^2(\omega)}{k_B \gamma^2(\omega)\omega^2 T} \\ \tau_B(\omega) &= \frac{d}{v_g(\omega)}, \end{aligned}\right\}, \tag{2.6}$$

and

where $\lambda$ is a constant which represents the physical properties of crystal structure. We use $\lambda = 1.9489$ in our work; $M$ is the total mass of the unit cell; $a$ is the lattice constant in the principal axis direction; $v_g$, $v_p$ are the phonon group velocity and the phase velocity, respectively; $\gamma$ is the mode Grüneisen parameter, and written as

$$\gamma = -\frac{V}{\omega(\vec{k})} \frac{\partial \omega(\vec{k})}{\partial V}, \tag{2.7}$$

$$v_p(\vec{k}) = \frac{\omega(\vec{k})}{\vec{k}} \tag{2.8}$$

and

$$v_g(\vec{k}) = \nabla_{\vec{k}} \omega(\vec{k}), \tag{2.9}$$

where $V$ is the cell volume, and the volume change of $\pm 4\%$ is considered in the calculation. $d$ is the effective size of the sample, which is considered to be 1 mm in this paper.

# 3. Results and discussion

The lattice constants and atomic positions of GeSe in the Pnma structure are optimized. The calculated lattice constants $a = 11.15$ Å, $b = 3.89$ Å, $c = 4.47$ Å are in good agreement with other theoretical values ($a = 11.17$ Å, $b = 3.88$ Å, $b = 4.52$ Å) [19]. Considering the overestimation of the lattice constant by geometric optimization under GGA approximation, the calculated lattice constants are slightly smaller than the experimental value [38]. The band structure of GeSe calculated along the high symmetry point of the first Brillouin zone is shown in figure 2. The black and red solid lines represent the calculation results of GGA-optB88 and HSE06 functional, respectively. GeSe in Pnma structure is an indirect band gap semiconductor. The VBM is at (0, 0, 0.35) along $\Gamma$–$Z$ path, and the CBM is at $\Gamma$ point. The band gap calculated by GGA-optB88 functional is 0.87 eV, which is consistent with 0.81 eV [39] and 0.85 eV [19] predicted by GGA-PBE functional. However, the fact that the excited states of the system are not considered in the calculation leads to the well-known underestimation of the semiconductor band gap by GGA functional. The band gap calculated by HSE06 functional is 1.34 eV. The calculated value is in good agreement with the latest experimental data (1.30 eV at 300 K) [40], so the description of the electronic structure of GeSe by HSE06 functional is more accurate.

The electron transport properties are closely related to the density of states and the distribution of charge density near the Fermi level. Figure 3a shows the isosurface distribution of band decomposition

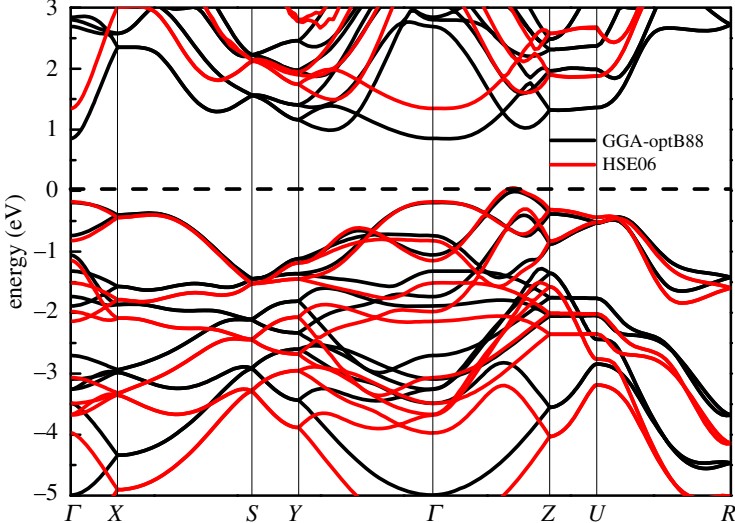

**Figure 2.** Band structure of GeSe.

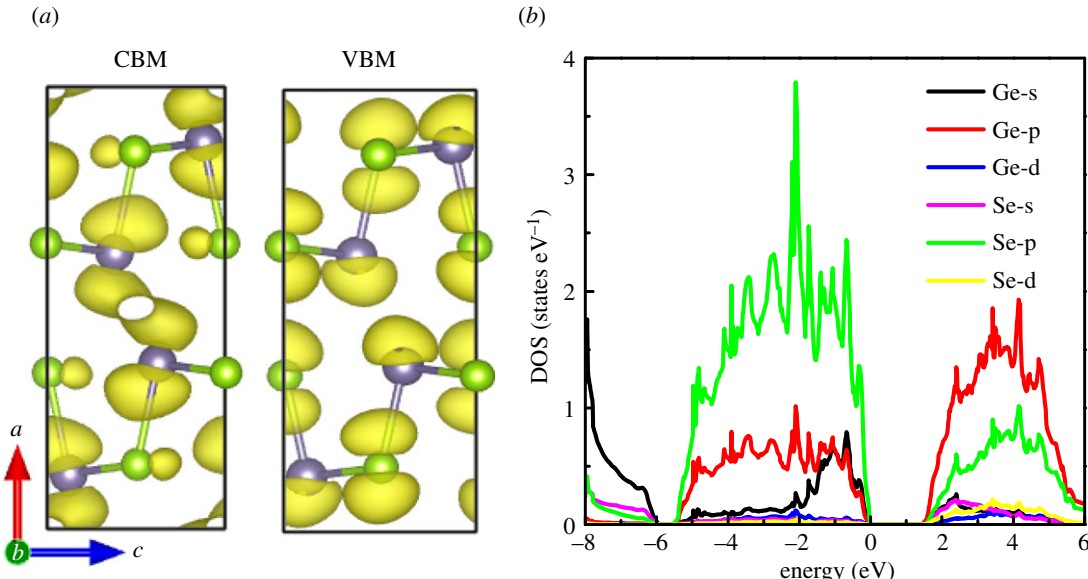

**Figure 3.** (*a*) Band decomposition charge density CBM and VBM (isosurface value is 0.003 e Å$^{-3}$) and (*b*) projected density of states.

charge density of VBM and CBM with charge density of 0.003 e Å$^{-3}$. The results show that, for CBM, the charge density connects two Ge atoms in the interlayer, which is beneficial to the electron transport, resulting in high mobility and electrical conductivity in the *a*-axis direction. Therefore, we infer that n-type GeSe has excellent electrical conductivity in the *a*-axis direction. On the contrary, for VBM, there is no charge density channel in the interlayer (*a*-axis direction), which prevents the vacancy transport, resulting in low electrical conductivity of p-type GeSe along the *a*-axis direction.

Figure 3*b* shows the projected density of states of GeSe near VBM and CBM. For the projected density of states, we first analyse that below the Fermi level (from −8.0 to −6.0 eV), which is mainly formed by the hybridization of Ge-s, Se-s and Se-p states. Due to the small energy level difference between Se-p and Ge-s states, Se-p and Ge-s states form strong bond in this energy range. The VBM in the higher energy range (−5.4 to −1.0 eV) is primarily dominated by the Ge-p and Se-p states. Near VBM (−1.0 to 0 eV), the DOS is mainly composed with the hybridization of Se-p, Ge-s and Ge-p states, especially Se-p state, which indicates Se atom plays an important role for VBM DOS near the Fermi level. That is to say, the electrical conductivity and Seebeck coefficient of p-type GeSe are mainly determined by Se-p state. Above the Fermi level (1.3 to 6.0 eV), near CBM, the projected state density is mainly composed of Ge-p and Se-p states, especially Ge-p state. This means that the electrical conductivity and Seebeck coefficient of n-type GeSe are mainly determined by the Ge-p states. Thus, we conclude that the

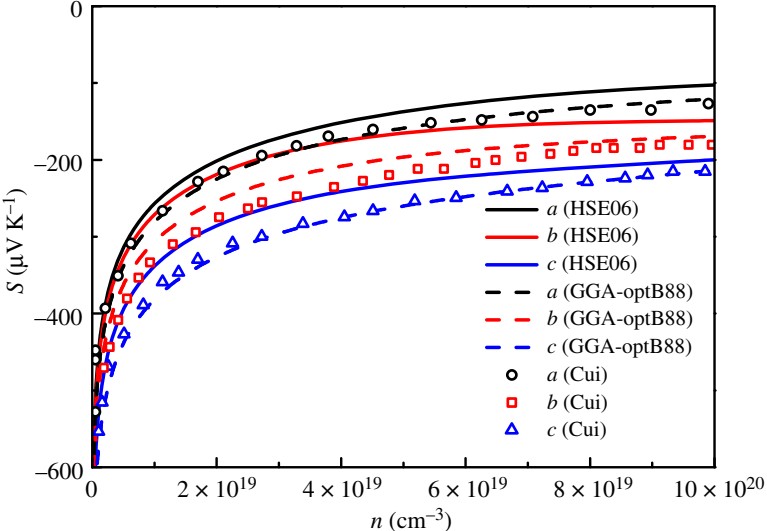

**Figure 4.** Variation of Seebeck coefficients with carrier concentration for n-type GeSe at 300 K.

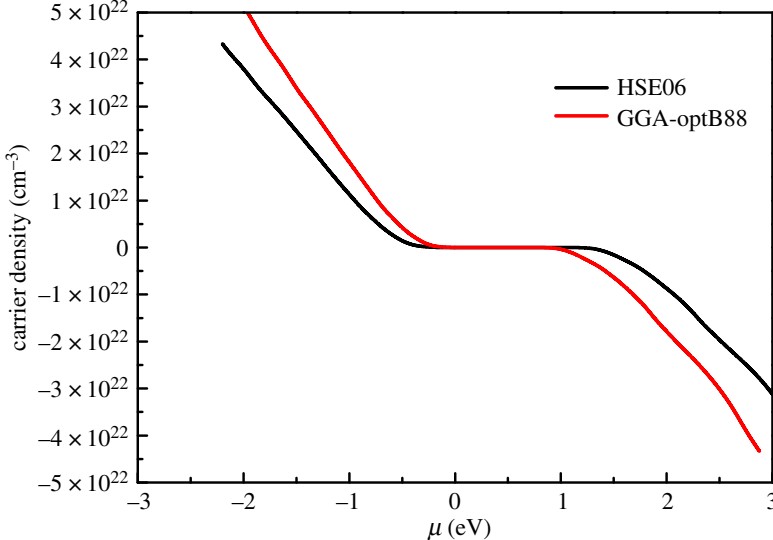

**Figure 5.** Carrier concentration as a function of chemical potential.

effective way to enhance the thermoelectric performance of p-type and n-type GeSe by adjusting the carrier concentration is adjusting the amount of Ge and Se atom, respectively. The results of Cui *et al.* [11] show that the thermoelectric performance of group VIIA elements replacing Se atom is superior to that of group VA elements replacing Ge atom for n-type GeSe. Though the DOS near the Fermi level for n-type GeSe is higher than that of p-type GeSe, the electronic conductivity for n-type GeSe is much larger due to the carrier transport pathway is formed along the *a*-axis direction.

Based on the energy eigenvalues calculated by HSE06 functional and GGA-optB88 functional, the Seebeck coefficients of n-type GeSe at 300 K along the *a*, *b* and *c* directions are shown in figure 4. The solid line is the result of HSE06 functional, the dashed line is the result of GGA-optB88 functional, and the scattered dots are the calculational results of GGA-PBE functional [11]. It can be seen from figure 4 that the calculated results by GGA-optB88 functional are in good agreement with other calculated results by GGA-PBE functional. At the same carrier concentration, the Seebeck coefficient calculated by HSE06 functional is larger than that calculated by GGA-optB88 functional; similar characteristics are also shown in the study of rock salt structure SrS [41].

To clarify the reason, figure 5 shows the relationship between carrier concentration and chemical potential. It can be clearly seen from figure 5 that the carrier concentration of GGA-optB88 functional is higher than that of HSE06 functional at the same chemical potential. This is due to that the doping

**Table 1.** Elastic constant $C$ (eV Å$^{-3}$), deformation potential constant $E_1$ (eV), effective mass $m^*$ ($m_e$), mobility $\mu$ (cm$^2$ V$^{-1}$ s$^{-1}$) and relaxation time $\tau$ (fs) at 300 K.

| | carrier type | $C$ (eV Å$^{-3}$) | $E_1$ (eV) | $m^*$ ($m_e$) | $\mu$ (cm$^2$ V$^{-1}$ s$^{-1}$) | $\tau$ (fs) |
|---|---|---|---|---|---|---|
| *a* | hole | 0.26 | −8.68 | 1.01 | 79.82 | 45.84 |
| | electron | 0.26 | −11.85 | 0.10 | 432.54 | 24.59 |
| *b* | hole | 0.29 | −11.41 | 0.52 | 100.07 | 29.59 |
| | electron | 0.29 | −3.33 | 1.59 | 384.24 | 347.36 |
| *c* | hole | 0.11 | −12.16 | 0.34 | 51.11 | 9.88 |
| | electron | 0.11 | −3.52 | 1.08 | 192.03 | 117.92 |

concentration in BoltzTraP software is determined by changing the chemical potential position from VBM up to above the CBM with rigid energy band model. Compared with the band structure calculated by the GGA-optB88 functional, the valence band of the HSE06 functional is higher, resulting in the doping concentration of HSE06 functional being lower than that of GGA-optB88 functional at the same chemical potential. Meanwhile, due to $S \propto n^{-3/2}$, the Seebeck coefficient calculated by HSE06 functional is larger than that calculated by GGA-optB88 functional at the same chemical potential, as shown in figure 4.

The electrical conductivity is calculated by Boltzmann transport theory within constant electron relaxation time approximation. In this paper, we use the deformation potential theory to estimate the electron relaxation time. It can be seen from figure 2 that the shape and variation trend of band structure along the k-point path calculated by HSE06 are similar to those calculated by GGA-optB88 functional. The parameters used to calculate the relaxation time in this work are calculated by GGA-optB88 functional. The elastic constant is calculated by taking into account five deformations from −2% to 2% with 1% intervals. Since the deformation potential constant is the change of band edge, smaller deformation should be considered. In this paper, five deformations of ±0.2%, 0.0% and ±0.5% are considered. The effective mass fitting is affected by the k-point interval and energy range. In our calculation, we use the same k-point interval (0.01 Å$^{-1}$) and energy range (0.15 eV above CBM for electron effective mass fitting and 0.1 eV below VBM for vacancy effective mass fitting). The fitting results of elastic constant, deformation potential constant and effective mass are shown in the electronic supplementary material. The calculated elastic constants, deformation potential constants, effective mass and relaxation time at 300 K are listed in table 1.

As listed in table 1, the elastic property displays anisotropy. The elastic constant along the *c* direction is smaller than that along the *a* and *b* directions, implying less elasticity along the *c* direction. Electronic supplementary material, figure S1 shows the variation in total energy versus uniaxial deformation. The variation of total energy versus strain shows that it is difficult to apply strain along the *c* direction. The feature is supported by the fact that the phase transition is easy to occur on the *c*-axis under pressure [42]. The deformation potential represents scattering of electron or hole by an acoustic phonon, which is related to the conduction band edge or valence band edge shift induced by longitudinal deformation.

The deformation potential depends not only on the direction but also on the carrier under consideration. The band energy of VBM and CBM with respect to lattice constant dilation are plotted in electronic supplementary material, figures S2 and S3. The band edge shift with lattice constant dilation fits perfectly linear, and the slope is deformation potential $E_1$. With respect to electron, it is intriguing that the absolute $E_1$ along the *a* direction is much larger than that along the *b* and *c* directions. This trend can be explained by the band decomposition charge density of CBM. The connected charge density in the interlayer makes the CB wave function more susceptible by the structural deformations with longitudinal phonons, thereby leading to absolute $E_1$ along the *a* direction larger than that along the *b* and *c* directions.

As can be seen from the band structure in figure 2, the band structure for CBM is so dispersed along the *a*-axis direction that the electron effective mass is very small (0.10$m_e$). Due to the smallest effective mass of electron along the *a*-axis direction, the electron mobility is largest in this direction (432.54 cm$^2$ V$^{-1}$ s$^{-1}$, at 300 K). The effective mass of electron is larger than that of vacancy along the *b*-axis, *c*-axis directions, while the absolute value of deformation potential constant of CBM is smaller than that of VBM, therefore, the mobility of n-type is larger than that of p-type along the *b*-axis and *c*-axis directions. Due to $\sigma \propto \mu$, the electrical conductivity of n-type is higher than that of p-type, which is consistent with the analysis of band decomposition charge density of VBM and CBM. For p-type

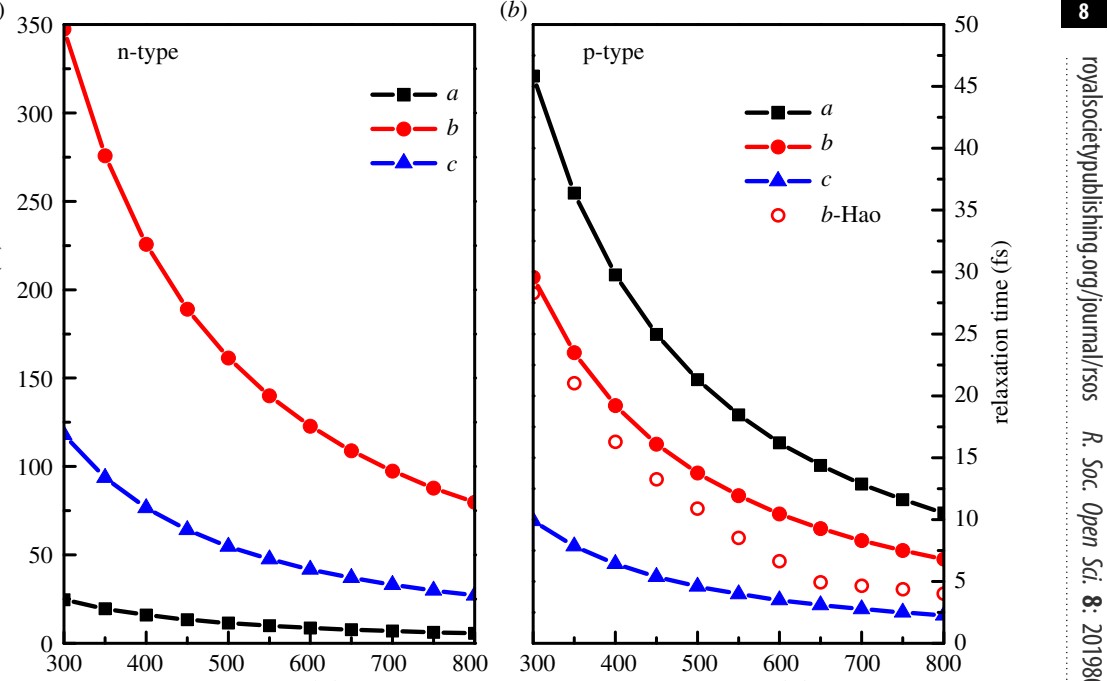

**Figure 6.** Relaxation time as function of temperature, hollow dots are the calculated value of Hao *et al.* [19].

GeSe, the mobility along the *b*-axis direction is larger than along the other two directions, which is beneficial to obtain high electrical transport performance at low carrier concentration along the *b*-axis direction. The carrier relaxation time is affected by deformation potential, the effective mass of state density and elastic constant. Using the calculated elastic constants, deformation potential constants and effective mass, we calculate the relaxation time at different temperature. The relation of relaxation time with temperature is shown in figure 6. The relaxation time of GeSe along the *b*-axis fitted by Hao *et al.* [19] are represented with hollow dots.

It is clearly seen from figure 6 that the carrier relaxation time decreases rapidly with temperature, which means that the higher the temperature, the stronger the scattering. The relaxation time for n-type is larger than that for p-type, which is consistent with the electron–phonon coupling calculation [43]. For n-type, the relaxation time in the *b*–*c* plane is significantly larger than that along the interlayer direction. For example, at 300 K, the relaxation time along the *b*-axis and *c*-axis directions are 347 and 118 fs, respectively, while the relaxation time is only 25 fs along the *a*-axis direction, this feature is also shown in the interlayer structure $SnSe_2$ [44].

We calculate the mode Grüneisen parameter, phonon group velocity, phase velocity and average free path of phonons along the *a*, *b* and *c* directions based on our previous work about phonon spectrum of GeSe [23]. The Umklapp scattering and boundary scattering are considered in our calculation. Figure 7 shows the calculated lattice thermal conductivity. The lattice thermal conductivity satisfies the approximate $1/T$ law with temperature. The lattice thermal conductivity along *a*, *b* and *c* directions is 1.11, 2.79 and $1.55\,\mathrm{W\,m^{-1}\,K^{-1}}$ at 300 K, and decreases to 0.70, 1.73 and $0.95\,\mathrm{W\,m^{-1}\,K^{-1}}$ at 800 K, respectively. The thermal conductivity of the *b*–*c* plane is larger than that of the interlayer direction, and the lattice thermal conductivity in the *b*-axis direction is the largest. The lattice thermal conductivity of Pnma structure SnSe also has similar characteristics [46]. The main reason for the above characteristics of lattice thermal conductivity is that the different phonon group velocities along different direction, as shown in previous research of phonon spectrums [23]. Similar to SnSe with the same Pnma structure, GeSe also has low lattice thermal conductivity. The strong anharmonicity and the van der Waals force in interlayer direction are beneficial to low thermal conductivity. It can be clearly seen from figure 7 that the calculated lattice thermal conductivity is much larger than that calculated by Hao *et al.* [19]. This is attributed to Hao *et al.* only considering the contribution of acoustic branches with the Debyee–Callaway model, but all acoustic and optical branches are considered in our work. It can be seen from figure 7 that the calculated average lattice thermal conductivity is in good agreement with the experimental data [9,21,45].

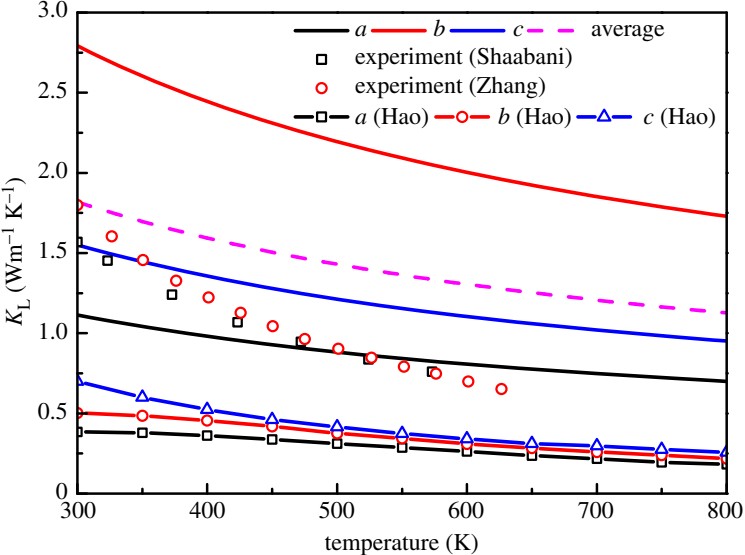

**Figure 7.** Lattice thermal conductivity along different directions, and compared with the experimental value [9,21,45] and theoretical calculation value [19].

The thermoelectric properties of GeSe are calculated using the energy eigenvalues calculated by HSE06 functional. The thermoelectric transport properties at 700 K are shown in figure 8.

It can be seen from figure 8$a$ that the Seebeck coefficient decreases with carrier concentration. In the range of carrier concentration from $1 \times 10^{18}$ to $1 \times 10^{20}$, the Seebeck coefficient of n-type GeSe is higher than that of p-type. The Seebeck coefficient of n-type shows obvious anisotropy. For n-type GeSe, the Seebeck coefficient along the $a$-axis direction is lower than that along the $b$-axis and $c$-axis directions. For n-type GeSe, along the $a$-axis direction, the absolute value of Seebeck coefficient is 596 µV K$^{-1}$ at $1 \times 10^{18}$ cm$^{-3}$, while it decreases to 220 µV K$^{-1}$ when the concentration increases to $1 \times 10^{20}$ cm$^{-3}$. High Seebeck coefficient means that GeSe may have considerable thermoelectric merit. By contrast to the Seebeck coefficient, the ratio of electrical conductivity to relaxation time ($\sigma/\tau$) increases with carrier concentration from $1 \times 10^{18}$ to $1 \times 10^{20}$ cm$^{-3}$. The $\sigma/\tau$ is very small when the carrier concentration is lower than $1 \times 10^{19}$ cm$^{-3}$. Therefore, increasing the carrier concentration is an important means to improve the electrical conductivity. Because n-type GeSe forms an electron transport channel along the $a$-axis direction, the $\sigma/\tau$ of n-type is much larger than that of p-type along the $a$-axis direction. For example, when the carrier concentration is $1 \times 10^{19}$ cm$^{-3}$, along the $a$-axis direction, the $\sigma/\tau$ is $2.26 \times 10^{18}$ S m$^{-1}$ s$^{-1}$ for n-type, while that is only $2.40 \times 10^{17}$ S m$^{-1}$ s$^{-1}$ for p-type. The $\sigma/\tau$ of n-type GeSe along the $a$-axis direction is much larger than that along the $b$-axis and $c$-axis directions. For example, when the carrier concentration is $1 \times 10^{19}$ cm$^{-3}$, the $\sigma/\tau$ of n-type GeSe along the $b$-axis direction is $3.06 \times 10^{17}$ S m$^{-1}$ s$^{-1}$, which is only 14% of that along the $a$-axis direction.

Using the calculated electron relaxation time $\tau$, the power factor PF = $S^2\sigma$ is calculated, as shown in figure 8$c$. The power factor of n-type GeSe is much larger than that of p-type GeSe. For n-type GeSe, the power factor along the $b$-axis direction is larger than that along the $a$-axis direction due to the larger electronic relaxation time $\tau$. For example, when the carrier concentration is $1 \times 10^{20}$ cm$^{-3}$, the power factor along the $b$-axis direction is 0.026 W m$^{-1}$ K$^{-2}$, which is 4 times that along the $a$-axis direction. Besides the power factor, the thermoelectric merit value is also related to the thermal conductivity $K$. The thermal conductivity includes the lattice thermal conductivity $K_l$ and the electronic thermal conductivity $K_e$. The electrical conductivity is proportional to the electronic thermal conductivity. According to Wiedemann–Franz law, the relation of electrical conductivity and electronic thermal conductivity is given as $K_e = \sigma LT$, where $T$ is the absolute temperature and $L$ is the Lorenz number. Using the calculated electrical conductivity $\sigma$, we calculate the electronic thermal conductivity. Combined with the calculated lattice thermal conductivity, the calculated total thermal conductivity is plotted in figure 8$d$. the lattice thermal conductivity is the main part of the thermal conductivity in low carrier concentration. With the increase of carrier concentration, the electronic thermal conductivity becomes more important, due to the conductivity increasing with the carrier concentration. The thermoelectric merit value ZT value of GeSe at 700 K is shown in figure 8$e$. In order to compare ZT value with polycrystalline sample, we consider the average ZT

Whoa

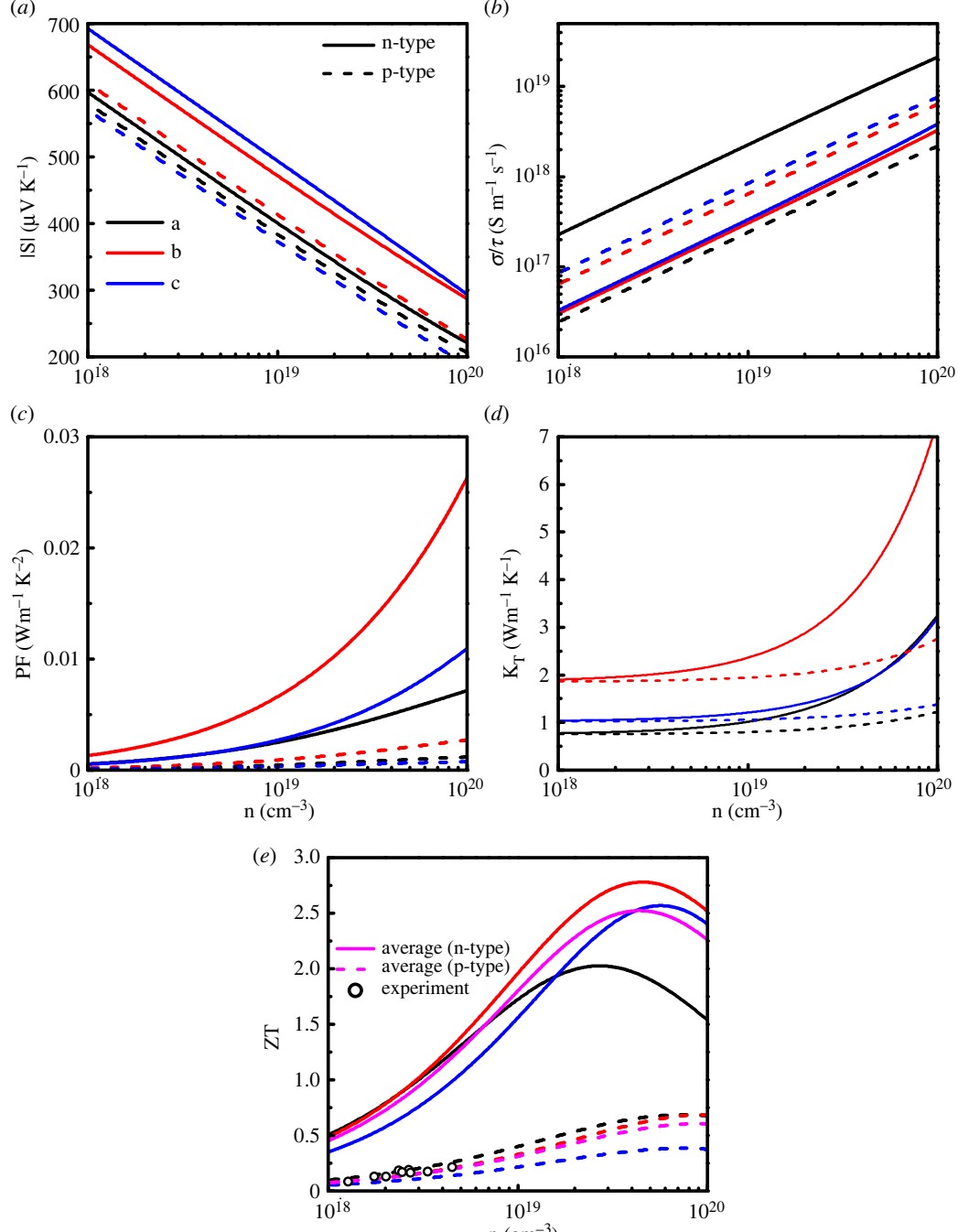

**Figure 8.** Variation of thermoelectric properties of GeSe with carrier concentration at 700 K, (*a*) Seebeck coefficient *S*, (*b*) electrical conductivities relative to relaxation time $\sigma/\tau$, (*c*) power factor PF, (*d*) thermal conductivity $K_T$ and (*e*) thermoelectric merit value ZT. The hollow dot in the figure is the experimental value [21].

value ($ZT_{ave} = PF_{ave}/K_{ave}$), where $PF_{ave}$ is average power factor ($PF_{ave} = (1/3)$ $(PF_a + PF_b + PF_c)$),$K_{ave}$ denotes average thermal conductivity ($K_{ave} = (1/3)$ $(K_a + K_b + K_c)$). It can be seen from figure 7*d* that the ZT value of n-type GeSe is much larger than that of p-type in the range of carrier concentration from $1 \times 10^{18}$ cm$^{-3}$ to $1 \times 10^{20}$ cm$^{-3}$. The calculated ZT values of p-type GeSe are in good agreement with the experimental values [21] in the low concentration region (approx. $10^{18}$ cm$^{-3}$) at 700 K. The ZT value of p-type GeSe is 0.1, while that of n-type is 0.5 at $1 \times 10^{18}$ cm$^{-3}$ carrier concentration, 700 K. The ZT value of p-type GeSe reaches 0.6 at $1 \times 10^{20}$ cm$^{-3}$ carrier concentration and 700 K, while the optimal ZT value of n-type is 2.5 at $4 \times 10^{19}$ cm$^{-3}$. The results show that n-type GeSe has excellent thermoelectric properties and is a promising thermoelectric material.

# 4. Conclusion

The electronic structure, band decomposition charge density and thermoelectric transport properties of GeSe are calculated by first-principles method and Boltzmann transport theory. The band gap calculated by HSE06 functional is 1.34 eV, which can be used to predict the electronic structure of GeSe. For p-type GeSe, there is no conductive pathway for carrier transport near the VBM, resulting in low electrical conductivity along the *a*-axis direction. By contrast, for n-type GeSe, a charge channel is formed along the *a*-axis direction near the CBM, which is favourable for electrical conductivity along the *a*-axis direction. The calculation results of lattice thermal conductivity show that the lattice thermal conductivity of GeSe along the *a*-axis direction is lower than that in the *b*–*c* plane. The lattice thermal conductivity along the *a*-axis direction of GeSe is 1.11 $Wm^{-1}K^{-1}$ at 300 K. The ZT value of n-type GeSe reaches the maximum value of 2.5 at 700 K, $4 \times 10^{19}$ $cm^{-3}$ carrier concentration, which is much higher than that of p-type GeSe at the same carrier concentration. The ZT value of p-type GeSe is 0.1 at 700 K, low concentration ($1 \times 10^{18}$ $cm^{-3}$), while that of n-type GeSe is 0.5, indicating that n-type GeSe is more favourable for thermoelectric properties than p-type GeSe.

Data accessibility. The article's supporting data can be accessed at the Dryad Digital Repository (https://doi.org/10.5061/dryad.1c59zw3t7) [47].

Authors' contributions. Q.F. carried out calculation, analysed data and drafted the manuscript. JY. conceived and designed this work. J.C and C.L. revised the draft. All authors approved the final version of the manuscript.

Competing interests. We declare we have no competing interests.

Funding. We would like to thank the financial supports from the Scientific Research Foundation of the Education Department of Sichuan Province (grant no. 18CZ0030), Science and Technology Bureau of Leshan City (grant nos. 19GZD006 and 20GZD033) and Leshan Normal University (grant nos. LZD022, LZDP014, DGZZ202013 and DGZZ202029).

Acknowledgements. We thank for the computing support of the Lvliang supercomputing centre.

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
