## [Peer Review File · Royal Society Open Science]

Review History

RSOS-201980.R0 (Original submission)

Review form: Reviewer 1

Is the manuscript scientifically sound in its present form?

Yes

Are the interpretations and conclusions justified by the results?

Yes

Is the language acceptable?

Yes

Do you have any ethical concerns with this paper?

No

Have you any concerns about statistical analyses in this paper?

No

Recommendation?

Accept with minor revision (please list in comments)

Comments to the Author(s)

The manuscript entitled "Thermoelectric performances for both p- and n- type GeSe" by Fan and co-workers discuss and investigate the electronic structure, band decomposition charge density and thermoelectric transport properties of both p- and n-type GeSe using first principles method and Boltzmann transport theory. For p-type GeSe, the electrical conductivity along a-direction is very low because the Se-p electrons near VBM pushes the Ge-p electron away from its interlayer a-axis direction. For n-type GeSe, however, due to the formation of a charge channel along the a-direction near CBM, electrical conductivity along a-direction is comparably higher. In addition to that, the lattice thermal conductivity of GeSe along a-axis direction is lower than that in b-c plane as well. As result, the thermoelectric figure of merit for n-type GeSe is calculated to be 2.5 at 700 K with $4 \times 10^{19} \text{ cm}^{-3}$ carrier concentration, which is much higher comparable to p-type GeSe with same the carrier concentration, indicating n-type GeSe can be more promising thermoelectric material than p-type. This work consists of electronic structure calculation with proper discussion of the thermoelectric properties of GeSe. I recommend the paper to go through. However, I think few information have to be added before publishing in Royal Society Open Science.

1. Authors have mentioned that, for n-type the relaxation time along b-c plane is significantly larger than along the interlayer direction. Authors need to discuss the reason behind it in detail.
2. Authors have shown the relaxation time of electron and holes in different crystallographic direction at 300 K in table 1. The relaxation time for electron is higher than hole in b and c-directions, however lower in a-direction. Authors need to explain about this opposite behavior.
3. In page 7, authors have written '...Hao et al. only consider contribution of acoustic branches with Debye-Callaway model, but all acoustic and optical branches are considered in our work...'. Therefore, authors need to show how much contribution have been obtained from the optical phonons to the heat propagation as in general the major part of the heat is mainly carried by acoustic phonons.
4. Authors must provide carrier concentration dependent total thermal conductivity at 700 K as well, as it has been used to determine the figure of merit of the studied material.
5. Additionally, I would request the authors to mention J. Am. Chem. Soc. 2020, 142, 12237, Angew. Chem. Int. Ed. 2018, 57,15167 and J. Mater. Chem. A, 2020, 8, 12226 as these are relevant to this work.

Review form: Reviewer 2

Is the manuscript scientifically sound in its present form?

No

Are the interpretations and conclusions justified by the results?

No

Is the language acceptable?

Yes

Do you have any ethical concerns with this paper?

No

Have you any concerns about statistical analyses in this paper?

No

Recommendation?

Major revision is needed (please make suggestions in comments)

Comments to the Author(s)

The authors investigate thermoelectric properties of p-type and n-type GeSe using first-principles calculation. Their important finding is that n-type GeSe can exhibit high thermoelectric performance, $ZT \sim 2.5$ at 700 K. Also they mention some characteristics of the electronic structure of GeSe.

Theoretical calculation looking for high ZT is important both in fundamental and applicational viewpoints, and this work offers a possible candidate on the basis of their careful calculations. However, I do not recommend publication of this manuscript at the present form. This is because I find several problematic descriptions that are scientifically not sound as follows:

(1) page 3, 3 lines below Fig. 1:

"In many theoretical calculations, the electron relaxation time is obtained by comparing the Seebeck coefficient at a certain temperature and carrier concentration with the experimental data."

I cannot understand this sentence. In the Boltzmann transport theory within the constant relaxation time approximation, the Seebeck coefficient is independent of the relaxation time. Therefore, it is not possible to determine the relaxation time from the Seebeck coefficient. Usually, the relaxation time is determined rather from the electrical resistivity.

(2) page 4, Eq. (6): Why do the authors use $\lambda = 1.9489$? How is this choice validated?

(3) page 4, Eqs. (6),(8)-(9): These equations seem peculiar. In the right hand side of Eq. (8), the numerator is a scalar but the denominator is a vector. How does the author define this quantity? In Eq. (9), v_a seems to be a vector quantity because the right hand side of Eq. (9) is a vector. However, in that case, how does one calculate the right hand side of Eq. (6)?

(4) page 5: The authors speculate the role of the lone pair electrons for the VBM state while no reasoning is presented. I do not think that a simple electron density and the band structure separately shown in Figs. 2 and 3 are not sufficient reasoning for it.

(5) page 5: It is naturally expected that the low-energy valence bands consist of the Se-p hybridized with the Ge-p (bonding state), and the low-energy conduction bands consist of the Ge-p hybridized with the Se-p (antibonding state). However, the authors say that "Near VBM (-1.0 eV to 0 eV), it is mainly the anti-bonding formed by Ge-p and Se-p states." and "Above the Fermi level (1.3 eV to 6.0 eV), the projected state density is mainly composed of Ge-p and Se-p bonding states." It is usually the case that the bonding state is energetically lower than the antibonding state, but the authors claim that this is not the case here. Please provide the reason why the authors think so. The authors also say "The anti-bond formed by Ge-p and Se-p states pushes charge away from a-axis direction, and the charge is easy to gather in the b-axis and c-axis directions, resulting in higher electrical conductivity in b-axis and c-axis direction", which I cannot understand the reasoning of this claim.

(6) page 5: I cannot find the calculation condition for the transport properties, in particular, the number of k-points. Usually, the number of k-mesh should be very fine for the transport calculations compared with the simple band structure calculation. Please check whether the calculated transport properties are unchanged by increasing the number of k-points.

(7) page 6: "Meanwhile, due to $S \propto n^{-3/2}$, the Seebeck coefficient calculated by HSE06 functional is larger than that calculated by GGA-optB88 functional at the same carrier concentration"

I cannot understand this sentence. If the authors would like to use a " $S \propto n^{-3/2}$ " relation, why does the Seebeck coefficient become different "at the same carrier concentration"?

(8) Fig. 5: Is the energy zero in this figure consistent with that shown in Fig.3? It seems that the energy zeros are different between these figures. Please provide the definition of the energy zero in Fig. 5.

(9) The relaxation times presented in Table 1 are partially too optimistic, reaching several hundreds of [fs], and this optimistic value is closely related to the main finding of this manuscript, i.e., the high ZT in the n-type GeSe. The author should carefully discuss the validity of their theoretical prediction because they use a rather simple approximation for estimating the relaxation time. For example, is it possible to compare their calculation and calculation presented in the study [39] raised by the authors as "consistent" while the details of consistency is not shown in the present manuscript?

(10) The authors estimate the electrical thermal conductivity using the Wiedemann Franz law, but the BoltzTraP code they employed in this study should give the calculated electrical thermal conductivity without using that law. I do not think that there is no need to use the additional approximation (Wiedemann Franz law) here.

Minor issues:

(1) Fig. 1: It seems that the crystal structures are shown using the VESTA software, which requires the appropriate reference while not shown in this manuscript. In the caption of Fig. 1, se -> Se.

(2) Figs. 4, 8: The unit of the Seebeck coefficient [mV/K] seems to be wrong.

Decision letter (RSOS-201980.R0)

Dear Professor Yang

The Editors assigned to your paper RSOS-201980 "Thermoelectric performances for both p- and n- type GeSe" have now received comments from reviewers and would like you to revise the paper in accordance with the reviewer comments and any comments from the Editors. Please note this decision does not guarantee eventual acceptance.

Please submit your revised manuscript and required files (see below) no later than 21 days from today's (ie 20-Jan-2021) date. Note: the ScholarOne system will 'lock' if submission of the revision is attempted 21 or more days after the deadline. If you do not think you will be able to meet this deadline please contact the editorial office immediately.

on behalf of Dr Talha Erdem (Associate Editor) and Miles Padgett (Subject Editor)
openscience@royalsociety.org

Associate Editor Comments to Author (Dr Talha Erdem):

Comments to the Author:

Dear Professor Yang,

We have now received the response from the reviewers. The reviewers, especially Reviewer 2, have issued important concerns regarding your manuscript. We kindly ask you to revise the manuscript in the light of these comments of the reviewers that are copied below:

Reviewer 1:

The manuscript entitled "Thermoelectric performances for both p- and n- type GeSe" by Fan and co-workers discuss and investigate the electronic structure, band decomposition charge density and thermoelectric transport properties of both p- and n-type GeSe using first principles method and Boltzmann transport theory. For p-type GeSe, the electrical conductivity along a-direction is very low because the Se-p electrons near VBM pushes the Ge-p electron away from its interlayer a-axis direction. For n-type GeSe, however, due to the formation of a charge channel along the a-direction near CBM, electrical conductivity along a-direction is comparably higher. In addition to that, the lattice thermal conductivity of GeSe along a-axis direction is lower than that in b-c plane as well. As result, the thermoelectric figure of merit for n-type GeSe is calculated to be 2.5 at 700 K with $4 \times 10^{19} \text{ cm}^{-3}$ carrier concentration, which is much higher comparable to p-type GeSe with same the carrier concentration, indicating n-type GeSe can be more promising thermoelectric material than p-type. This work consists of electronic structure calculation with proper discussion of the thermoelectric properties of GeSe. I recommend the paper to go through. However, I think few information have to be added before publishing in Royal Society Open Science.

1. Authors have mentioned that, for n-type the relaxation time along b-c plane is significantly larger than along the interlayer direction. Authors need to discuss the reason behind it in detail.
2. Authors have shown the relaxation time of electron and holes in different crystallographic direction at 300 K in table 1. The relaxation time for electron is higher than hole in b and c-directions, however lower in a-direction. Authors need to explain about this opposite behavior.

3. In page 7, authors have written ‘...Hao et al. only consider contribution of acoustic branches with Debye-Callaway model, but all acoustic and optical branches are considered in our work...’. Therefore, authors need to show how much contribution have been obtained from the optical phonons to the heat propagation as in general the major part of the heat is mainly carried by acoustic phonons.

4. Authors must provide carrier concentration dependent total thermal conductivity at 700 K as well, as it has been used to determine the figure of merit of the studied material.

5. Additionally, I would request the authors to mention J. Am. Chem. Soc. 2020, 142, 12237, Angew. Chem. Int. Ed. 2018, 57,15167 and J. Mater. Chem. A, 2020, 8, 12226 as these are relevant to this work.

Reviewer 2:

The authors investigate thermoelectric properties of p-type and n-type GeSe using first-principles calculation. Their important finding is that n-type GeSe can exhibit high thermoelectric performance, $ZT \sim 2.5$ at 700 K. Also they mention some characteristics of the electronic structure of GeSe.

Theoretical calculation looking for high ZT is important both in fundamental and applicational viewpoints, and this work offers a possible candidate on the basis of their careful calculations. However, I do not recommend publication of this manuscript at the present form. This is because I find several problematic descriptions that are scientifically not sound as follows:

(1) page 3, 3 lines below Fig. 1:

"In many theoretical calculations, the electron relaxation time is obtained by comparing the Seebeck coefficient at a certain temperature and carrier concentration with the experimental data."

I cannot understand this sentence. In the Boltzmann transport theory within the constant relaxation time approximation, the Seebeck coefficient is independent of the relaxation time. Therefore, it is not possible to determine the relaxation time from the Seebeck coefficient. Usually, the relaxation time is determined rather from the electrical resistivity.

(2) page 4, Eq. (6): Why do the authors use $\lambda = 1.9489$? How is this choice validated?

(3) page 4, Eqs. (6),(8)-(9): These equations seem peculiar. In the right hand side of Eq. (8), the numerator is a scalar but the denominator is a vector. How does the author define this quantity? In Eq. (9), v_a seems to be a vector quantity because the right hand side of Eq. (9) is a vector. However, in that case, how does one calculate the right hand side of Eq. (6)?

(4) page 5: The authors speculate the role of the lone pair electrons for the VBM state while no reasoning is presented. I do not think that a simple electron density and the band structure separately shown in Figs. 2 and 3 are not sufficient reasoning for it.

(5) page 5: It is naturally expected that the low-energy valence bands consist of the Se-p hybridized with the Ge-p (bonding state), and the low-energy conduction bands consist of the Ge-p hybridized with the Se-p (antibonding state). However, the authors say that "Near VBM (-1.0 eV to 0 eV), it is mainly the anti-bonding formed by Ge-p and Se-p states." and "Above the Fermi level (1.3 eV to 6.0 eV), the projected state density is mainly composed of Ge-p and Se-p bonding states." It is usually the case that the bonding state is energetically lower than the antibonding state, but the authors claim that this is not the case here. Please provide the reason why the authors think so. The authors also say "The anti-bond formed by Ge-p and Se-p states pushes charge away from a-axis direction, and the charge is easy to gather in the b-axis and c-axis

directions, resulting in higher electrical conductivity in b-axis and c-axis direction", which I cannot understand the reasoning of this claim.

(6) page 5: I cannot find the calculation condition for the transport properties, in particular, the number of k-points. Usually, the number of k-mesh should be very fine for the transport calculations compared with the simple band structure calculation. Please check whether the calculated transport properties are unchanged by increasing the number of k-points.

(7) page 6: "Meanwhile, due to $S \propto n^{-3/2}$, the Seebeck coefficient calculated by HSE06 functional is larger than that calculated by GGA-optB88 functional at the same carrier concentration"

I cannot understand this sentence. If the authors would like to use a " $S \propto n^{-3/2}$ " relation, why does the Seebeck coefficient become different "at the same carrier concentration"?

(8) Fig. 5: Is the energy zero in this figure consistent with that shown in Fig.3? It seems that the energy zeros are different between these figures. Please provide the definition of the energy zero in Fig. 5.

(9) The relaxation times presented in Table 1 are partially too optimistic, reaching several hundreds of [fs], and this optimistic value is closely related to the main finding of this manuscript, i.e., the high ZT in the n-type GeSe. The author should carefully discuss the validity of their theoretical prediction because they use a rather simple approximation for estimating the relaxation time. For example, is it possible to compare their calculation and calculation presented in the study [39] raised by the authors as "consistent" while the details of consistency is not shown in the present manuscript?

(10) The authors estimate the electrical thermal conductivity using the Wiedemann Franz law, but the BoltzTraP code they employed in this study should give the calculated electrical thermal conductivity without using that law. I do not think that there is no need to use the additional approximation (Wiedemann Franz law) here.

Minor issues:

(1) Fig. 1: It seems that the crystal structures are shown using the VESTA software, which requires the appropriate reference while not shown in this manuscript. In the caption of Fig. 1, se -> Se.

(2) Figs. 4, 8: The unit of the Seebeck coefficient [mV/K] seems to be wrong.

Reviewer comments to Author:

Reviewer: 1

Comments to the Author(s)

The manuscript entitled "Thermoelectric performances for both p- and n- type GeSe" by Fan and co-workers discuss and investigate the electronic structure, band decomposition charge density and thermoelectric transport properties of both p- and n-type GeSe using first principles method and Boltzmann transport theory. For p-type GeSe, the electrical conductivity along a-direction is very low because the Se-p electrons near VBM pushes the Ge-p electron away from its interlayer a-axis direction. For n-type GeSe, however, due to the formation of a charge channel along the a-direction near CBM, electrical conductivity along a-direction is comparably higher. In addition to that, the lattice thermal conductivity of GeSe along a-axis direction is lower than that in b-c plane as well. As result, the thermoelectric figure of merit for n-type GeSe is calculated to be 2.5 at 700 K with $4 \times 10^{19} \text{ cm}^{-3}$ carrier concentration, which is much higher comparable to p-type GeSe with

same the carrier concentration, indicating n-type GeSe can be more promising thermoelectric material than p-type. This work consists of electronic structure calculation with proper discussion of the thermoelectric properties of GeSe. I recommend the paper to go through. However, I think few information have to be added before publishing in Royal Society Open Science.

1. Authors have mentioned that, for n-type the relaxation time along b-c plane is significantly larger than along the interlayer direction. Authors need to discuss the reason behind it in detail.
2. Authors have shown the relaxation time of electron and holes in different crystallographic direction at 300 K in table 1. The relaxation time for electron is higher than hole in b and c-directions, however lower in a-direction. Authors need to explain about this opposite behavior.
3. In page 7, authors have written '...Hao et al. only consider contribution of acoustic branches with Debye-Callaway model, but all acoustic and optical branches are considered in our work...'. Therefore, authors need to show how much contribution have been obtained from the optical phonons to the heat propagation as in general the major part of the heat is mainly carried by acoustic phonons.
4. Authors must provide carrier concentration dependent total thermal conductivity at 700 K as well, as it has been used to determine the figure of merit of the studied material.
5. Additionally, I would request the authors to mention J. Am. Chem. Soc. 2020, 142, 12237, Angew. Chem. Int. Ed. 2018, 57,15167 and J. Mater. Chem. A, 2020, 8, 12226 as these are relevant to this work.

Reviewer: 2

Comments to the Author(s)

The authors investigate thermoelectric properties of p-type and n-type GeSe using first-principles calculation. Their important finding is that n-type GeSe can exhibit high thermoelectric performance, $ZT \sim 2.5$ at 700 K. Also they mention some characteristics of the electronic structure of GeSe.

Theoretical calculation looking for high ZT is important both in fundamental and applicational viewpoints, and this work offers a possible candidate on the basis of their careful calculations. However, I do not recommend publication of this manuscript at the present form. This is because I find several problematic descriptions that are scientifically not sound as follows:

(1) page 3, 3 lines below Fig. 1:

"In many theoretical calculations, the electron relaxation time is obtained by comparing the Seebeck coefficient at a certain temperature and carrier concentration with the experimental data."

I cannot understand this sentence. In the Boltzmann transport theory within the constant relaxation time approximation, the Seebeck coefficient is independent of the relaxation time. Therefore, it is not possible to determine the relaxation time from the Seebeck coefficient. Usually, the relaxation time is determined rather from the electrical resistivity.

(2) page 4, Eq. (6): Why do the authors use $\lambda = 1.9489$? How is this choice validated?

(3) page 4, Eqs. (6),(8)-(9): These equations seem peculiar. In the right hand side of Eq. (8), the numerator is a scalar but the denominator is a vector. How does the author define this quantity? In Eq. (9), v_a seems to be a vector quantity because the right hand side of Eq. (9) is a vector. However, in that case, how does one calculate the right hand side of Eq. (6)?

(4) page 5: The authors speculate the role of the lone pair electrons for the VBM state while no reasoning is presented. I do not think that a simple electron density and the band structure separately shown in Figs. 2 and 3 are not sufficient reasoning for it.

(5) page 5: It is naturally expected that the low-energy valence bands consist of the Se-p hybridized with the Ge-p (bonding state), and the low-energy conduction bands consist of the Ge-p hybridized with the Se-p (antibonding state). However, the authors say that "Near VBM (-1.0 eV to 0 eV), it is mainly the anti-bonding formed by Ge-p and Se-p states." and "Above the Fermi level (1.3 eV to 6.0 eV), the projected state density is mainly composed of Ge-p and Se-p bonding states." It is usually the case that the bonding state is energetically lower than the antibonding state, but the authors claim that this is not the case here. Please provide the reason why the authors think so. The authors also say "The anti-bond formed by Ge-p and Se-p states pushes charge away from a-axis direction, and the charge is easy to gather in the b-axis and c-axis directions, resulting in higher electrical conductivity in b-axis and c-axis direction", which I cannot understand the reasoning of this claim.

(6) page 5: I cannot find the calculation condition for the transport properties, in particular, the number of k-points. Usually, the number of k-mesh should be very fine for the transport calculations compared with the simple band structure calculation. Please check whether the calculated transport properties are unchanged by increasing the number of k-points.

(7) page 6: "Meanwhile, due to $S \propto n^{-3/2}$, the Seebeck coefficient calculated by HSE06 functional is larger than that calculated by GGA-optB88 functional at the same carrier concentration"

I cannot understand this sentence. If the authors would like to use a " $S \propto n^{-3/2}$ " relation, why does the Seebeck coefficient become different "at the same carrier concentration"?

(8) Fig. 5: Is the energy zero in this figure consistent with that shown in Fig.3? It seems that the energy zeros are different between these figures. Please provide the definition of the energy zero in Fig. 5.

(9) The relaxation times presented in Table 1 are partially too optimistic, reaching several hundreds of [fs], and this optimistic value is closely related to the main finding of this manuscript, i.e., the high ZT in the n-type GeSe. The author should carefully discuss the validity of their theoretical prediction because they use a rather simple approximation for estimating the relaxation time. For example, is it possible to compare their calculation and calculation presented in the study [39] raised by the authors as "consistent" while the details of consistency is not shown in the present manuscript?

(10) The authors estimate the electrical thermal conductivity using the Wiedemann Franz law, but the BoltzTraP code they employed in this study should give the calculated electrical thermal conductivity without using that law. I do not think that there is no need to use the additional approximation (Wiedemann Franz law) here.

Minor issues:

(1) Fig. 1: It seems that the crystal structures are shown using the VESTA software, which requires the appropriate reference while not shown in this manuscript. In the caption of Fig. 1, se -> Se.

(2) Figs. 4, 8: The unit of the Seebeck coefficient [mV/K] seems to be wrong.

===PREPARING YOUR MANUSCRIPT===

one version identifying all the changes that have been made (for instance, in coloured highlight, in bold text, or tracked changes);
 a 'clean' version of the new manuscript that incorporates the changes made, but does not highlight them. This version will be used for typesetting if your manuscript is accepted.

===PREPARING YOUR REVISION IN SCHOLARONE===

- Any electronic supplementary material (ESM).
- If you are requesting a discretionary waiver for the article processing charge, the waiver form must be included at this step.
- If you are providing image files for potential cover images, please upload these at this step, and inform the editorial office you have done so. You must hold the copyright to any image provided.
- A copy of your point-by-point response to referees and Editors. This will expedite the preparation of your proof.

- Ensure that your data access statement meets the requirements at <https://royalsociety.org/journals/authors/author-guidelines/#data>. You should ensure that you cite the dataset in your reference list. If you have deposited data etc in the Dryad repository, please include both the 'For publication' link and 'For review' link at this stage.
- If you are requesting an article processing charge waiver, you must select the relevant waiver option (if requesting a discretionary waiver, the form should have been uploaded at Step 3 'File upload' above).
- If you have uploaded ESM files, please ensure you follow the guidance at <https://royalsociety.org/journals/authors/author-guidelines/#supplementary-material> to include a suitable title and informative caption. An example of appropriate titling and captioning may be found at https://figshare.com/articles/Table_S2_from_Is_there_a_trade-off_between_peak_performance_and_performance_breadth_across_temperatures_for_aerobic_scope_in_teleost_fishes_/3843624.

Author's Response to Decision Letter for (RSOS-201980.R0)

See Appendices A & B.

RSOS-201980.R1 (Revision)

Review form: Reviewer 2

Is the manuscript scientifically sound in its present form?

No

Are the interpretations and conclusions justified by the results?

No

Is the language acceptable?

Yes

Do you have any ethical concerns with this paper?

No

Have you any concerns about statistical analyses in this paper?

No

Recommendation?

Accept with minor revision (please list in comments)

Comments to the Author(s)

The authors have responded to the referees' comments, and the manuscript is improved. However, I consider that the following point need to be addressed.

In the response 5, the authors said that low-energy valence bands (-5.4 eV to -1.0 eV) are the bonding states of the Ge-p and Se-p orbitals, and near VMC and low-energy conduction bands are anti-bonding states of them. I do not think this explanation is correct.

In GeSe, there are six Ge-p orbitals and six Se-p orbitals per formula unit (including the spin multiplicity). Thus, the number of the bonding states should be six and that for the anti-bonding states are also six. Considering the number of valence electrons in this system, the bonding states are fully occupied and the antibonding states are all empty. This is the reason why this system has a band gap. This situation can be found in many insulators. The authors claimed that the electronic states near VBM is also anti-bonding, which should be incorrect.

It seems that the authors consider the VBM state is anti-bonding on the basis of the charge density plot. However, it is clearly insufficient because what characterizes the anti-bonding state is not the density (absolute value of the wave function) but the phase of the wave function. And even for the bonding state, the Bloch phase factor and the complicated cancellation among the transfer integrals can induce quantum interference of the wave function, which results in a decrease of the density. Thus, a careful discussion is needed if one discusses the bonding or anti-bonding nature of the wave function on the basis of the wave-function or density plot.

Decision letter (RSOS-201980.R1)

Dear Professor Yang

On behalf of the Editors, we are pleased to inform you that your Manuscript RSOS-201980.R1 "Thermoelectric performances for both p- and n- type GeSe" has been accepted for publication in Royal Society Open Science subject to minor revision in accordance with the referees' reports. Please find the referees' comments along with any feedback from the Editors below my signature.

Please submit your revised manuscript and required files (see below) no later than 7 days from today's (ie 28-Apr-2021) date. Note: the ScholarOne system will 'lock' if submission of the revision is attempted 7 or more days after the deadline. If you do not think you will be able to meet this deadline please contact the editorial office immediately.

on behalf of Dr Talha Erdem (Associate Editor) and Miles Padgett (Subject Editor)
openscience@royalsociety.org

Associate Editor Comments to Author (Dr Talha Erdem):

Associate Editor: 1

Comments to the Author:

Dear Professor Yang,

The manuscript has now been reviewed by the referees. In the light of the reviewer comments, we believe some minor revisions are still necessary prior to the final acceptance decision. The reviewer comments can now be found below.

Regards,
Dr. Talha Erdem
Associate Editor

Reviewer's Comments:

The authors have responded to the referees' comments, and the manuscript is improved. However, I consider that the following point need to be addressed.

In the response 5, the authors said that low-energy valence bands (-5.4 eV to -1.0 eV) are the bonding states of the Ge-p and Se-p orbitals, and near VMC and low-energy conduction bands are anti-bonding states of them. I do not think this explanation is correct.

In GeSe, there are six Ge-p orbitals and six Se-p orbitals per formula unit (including the spin multiplicity). Thus, the number of the bonding states should be six and that for the anti-bonding states are also six. Considering the number of valence electrons in this system, the bonding states are fully occupied and the antibonding states are all empty. This is the reason why this system has a band gap. This situation can be found in many insulators. The authors claimed that the electronic states near VBM is also anti-bonding, which should be incorrect.

It seems that the authors consider the VBM state is anti-bonding on the basis of the charge density plot. However, it is clearly insufficient because what characterizes the anti-bonding state is not the density (absolute value of the wave function) but the phase of the wave function. And even for the bonding state, the Bloch phase factor and the complicated cancellation among the transfer integrals can induce quantum interference of the wave function, which results in a decrease of the density. Thus, a careful discussion is needed if one discusses the bonding or anti-bonding nature of the wave function on the basis of the wave-function or density plot.

Reviewer comments to Author:

Reviewer: 2

Comments to the Author(s)

The authors have responded to the referees' comments, and the manuscript is improved.

However, I consider that the following point need to be addressed.

In the response 5, the authors said that low-energy valence bands (-5.4 eV to -1.0 eV) are the bonding states of the Ge-p and Se-p orbitals, and near VMC and low-energy conduction bands are anti-bonding states of them. I do not think this explanation is correct.

In GeSe, there are six Ge-p orbitals and six Se-p orbitals per formula unit (including the spin multiplicity). Thus, the number of the bonding states should be six and that for the anti-bonding states are also six. Considering the number of valence electrons in this system, the bonding states are fully occupied and the antibonding states are all empty. This is the reason why this system has a band gap. This situation can be found in many insulators. The authors claimed that the electronic states near VBM is also anti-bonding, which should be incorrect.

It seems that the authors consider the VBM state is anti-bonding on the basis of the charge density plot. However, it is clearly insufficient because what characterizes the anti-bonding state is not the density (absolute value of the wave function) but the phase of the wave function. And even for the bonding state, the Bloch phase factor and the complicated cancellation among the transfer integrals can induce quantum interference of the wave function, which results in a decrease of the density. Thus, a careful discussion is needed if one discusses the bonding or anti-bonding nature of the wave function on the basis of the wave-function or density plot.

===PREPARING YOUR MANUSCRIPT===

While not essential, it will speed up the preparation of your manuscript proof if you format your references/bibliography in Vancouver style (please see

<https://royalsociety.org/journals/authors/author-guidelines/#formatting>). You should include DOIs for as many of the references as possible.

===PREPARING YOUR REVISION IN SCHOLARONE===

<https://royalsociety.org/journals/authors/author-guidelines/#data>. You should ensure that you cite the dataset in your reference list. If you have deposited data etc in the Dryad repository,

please only include the 'For publication' link at this stage. You should remove the 'For review' link.

Author's Response to Decision Letter for (RSOS-201980.R1)

See Appendix C.

Decision letter (RSOS-201980.R2)

Dear Professor Yang,

It is a pleasure to accept your manuscript entitled "Thermoelectric performances for both p- and n- type GeSe" in its current form for publication in Royal Society Open Science. The comments of the reviewer(s) who reviewed your manuscript are included at the foot of this letter.

on behalf of Dr Talha Erdem (Associate Editor) and Miles Padgett (Subject Editor)
openscience@royalsociety.org

Associate Editor Comments to Author (Dr Talha Erdem):
Comments to the Author:
Dear Prof. Yang,

Now that you addressed all the concerns of Reviewer 2, I suggest the acceptance of the manuscript in this current form.

Kind wishes,
Dr. Talha Erdem

Appendix A

Response to Reviewer 1

Thank you very much for your constructive suggestions. The advices are very useful for the amendment of our paper as well as for our future research.

We respond to the comments in the following point by point.

1. Authors have mentioned that, for n-type the relaxation time along b-c plane is significantly larger than along the interlayer direction. Authors need to discuss the reason behind it in detail.

Response:

The electron relaxation time is decided by elastic constant C , deformation potential E_1 based on the deformation potential theory. With respect to electron, the considerable deformation value is responsible for the larger electron relaxation time along “a direction”. We analyzed those parameters in addition to the electron relaxation time in the revised manuscript.

2. Authors have shown the relaxation time of electron and holes in different crystallographic direction at 300 K in table 1. The relaxation time for electron is higher than hole in b and c-directions, however lower in a-direction. Authors need to explain about this opposite behavior.

Response:

The relaxation time is not dependent on direction, but on carrier consideration. The response to this question is the same as the previous one.

3. In page 7, authors have written ‘...Hao et al. only consider contribution of acoustic branches with Debye-Callaway model, but all acoustic and optical

branches are considered in our work...'. Therefore, authors need to show how much contribution have been obtained from the optical phonons to the heat propagation as in general the major part of the heat is mainly carried by acoustic phonons.

Response:

In Hao's work, they explicitly mentioned that the total lattice thermal conductivity is written as a sum over one longitudinal and two transverse acoustic phonon branches. We cannot judge arbitrarily that the difference between the calculated thermal conductivity and the thermal conductivity from Hao et al. is the contribution of the optical branch, due to different semi-classical methods used.

4. Authors must provide carrier concentration dependent total thermal conductivity at 700 K as well, as it has been used to determine the figure of merit of the studied material.

Response:

In fact, the thermal conductivity is indeed a factor determining the ZT value.

We plotted the total thermal conductivity as a function of carrier concentration in the revised manuscript.

5. Additionally, I would request the authors to mention J. Am. Chem. Soc. 2020, 142, 12237, Angew. Chem. Int. Ed. 2018, 57,15167 and J. Mater. Chem. A, 2020, 8, 12226 as these are relevant to this work.

Response:

Thank you very much for mentioning those so important relevant articles. We mentioned them the revised manuscript.

Appendix B

Response to Reviewer 2

Thank you very much for your constructive suggestions and objective evaluation about our manuscript. Your guidance is very helpful for us to understand many basic problems. Your rigorous academic attitude will play an important guiding role in our future research work. We have studied the comments carefully and respond to the comments in the following point by point.

1. page 3, 3 lines below Fig. 1:

"In many theoretical calculations, the electron relaxation time is obtained by comparing the Seebeck coefficient at a certain temperature and carrier concentration with the experimental data."

I cannot understand this sentence. In the Boltzmann transport theory within the constant relaxation time approximation, the Seebeck coefficient is independent of the relaxation time. Therefore, it is not possible to determine the relaxation time from the Seebeck coefficient. Usually, the relaxation time is determined rather from the electrical resistivity.

Response:

As you mentioned, the Seebeck coefficient is independent of relaxation in the Boltzmann transport theory within the constant relaxation time approximation time. Maybe, the confusing description in manuscript leading your misunderstanding, I'll make corresponding explanation and

modification for the misunderstanding.

Usually, the relaxation time is a function of carrier concentration and temperature. With the Boltzmann transport theory, only the ratio of electrical conductivity to relaxation time (σ/τ) can be calculated, but not the electrical conductivity (σ). By comparing the calculated σ/τ value with the experimental σ value, τ can be obtained. However, in experiments usually give the function of electrical conductivity or electrical resistivity, Seebeck coefficient with temperature, but the corresponding carrier concentration is not given [1]. In this case, the calculated Seebeck coefficient at a certain temperature is usually compared with the experimental measurement at the same temperature to determine the carrier concentration. In order to avoid confusion, we revised it in revised manuscript.

2. page 4, Eq. (6): Why do the authors use $\lambda = 1.9489$? How is this choice validated?

Response: In our work, we use semi-classical approximation methods to calculate lattice thermal conductivity based on phonon spectrum. We use the expression of Umklapp scattering (Eq. (6)) from reference [2], the λ is the coefficient of the expression of Umklapp scattering (Eq. (6)) in reference [2], that is $\lambda = \frac{(6\pi^2)^{1/3}}{2} = 1.9489$.

3. page 4, Eqs. (6),(8)-(9): These equations seem peculiar. In the right hand side of Eq. (8), the numerator is a scalar but the denominator is a vector. How does the author define this quantity? In Eq. (9), v_a seems to be a vector quantity because the right hand side of Eq. (9) is a vector. However, in that case, how does one calculate the right hand side of Eq. (6)?

Response:

Thank you for pointing out the problem that should be avoided.

We are also puzzled why Eqs. (8) and (9) have the problem of vector scalar, because it is inconsistent with the display in our word document.

We speculate that the error occurred when word document was converted to PDF. We are sorry that we did not check it out when we confirmed the contribution. We have reedited these equations in revised manuscript.

We want to explain how to calculate the right side of Eq. (6). As mentioned in manuscript, when we calculate the thermal conductivity, we consider a finite number of k points in the direction of the principal axis. In other words, in the actual application, we calculate thermal conductivity along a certain principal axis direction.

4. page 5: The authors speculate the role of the lone pair electrons for the VBM state while no reasoning is presented. I do not think that a simple electron density and the band structure separately shown in Figs. 2

and 3 are not sufficient reasoning for it.

Response:

We admit that the concept of “lone pair electrons” is somewhat uncertain here. It is more appropriate to analyze from the charge density distribution. We have made corresponding changes in the revised draft.

5. page 5: It is naturally expected that the low-energy valence bands consist of the Se-p hybridized with the Ge-p (bonding state), and the low-energy conduction bands consist of the Ge-p hybridized with the Se-p (antibonding state). However, the authors say that "Near VBM (-1.0 eV to 0 eV), it is mainly the anti-bonding formed by Ge-p and Se-p states." and "Above the Fermi level (1.3 eV to 6.0 eV), the projected state density is mainly composed of Ge-p and Se-p bonding states." It is usually the case that the bonding state is energetically lower than the antibonding state, but the authors claim that this is not the case here. Please provide the reason why the authors think so. The authors also say "The anti-bond formed by Ge-p and Se-p states pushes charge away from a-axis direction, and the charge is easy to gather in the b-axis and c-axis directions, resulting in higher electrical conductivity in b-axis and c-axis direction", which I cannot understand the reasoning of this claim.

Response:

The bonding formed by Ge-p and Se-p states is located in low-energy VB

(- 5.4 eV to - 1.0 eV), and near VBM and low-energy CB is anti-bonding of Ge-p and Se-p states. "Above the Fermi level (1.3 eV to 6.0 eV), the projected state density is mainly composed of Ge-p and Se-p bonding states." Indeed, in the range of (1.3 eV to 6.0 eV) should be an anti-bonding state rather than a bonding state of Ge-p and Se-p.

About "The anti-bond formedin b-axis and c-axis direction ", As can be seen from the band decomposition charge density for VBM below, we find the Ge charge density are pushed away along intralayer a direction due to the anti-bonding formed by Ge-p and Se-p states, which prevents the holes transport along the intralayer a direction. The electrons pushed away from a direction relatively increase the charge distribution along the intralayer b and c directions. To better display this characteristic, we display band decomposed charge density with isosurface value 0.003 e / Å³. The modify was represented in the revised manuscript.

6. page 5: I cannot find the calculation condition for the transport properties, in particular, the number of k-points. Usually, the number of

k-mesh should be very fine for the transport calculations compared with the simple band structure calculation. Please check whether the calculated transport properties are unchanged by increasing the number of k-points.

Response:

Before using HSE06 functional, we have used GGA-optB88 to test the cut-off energy, total energy and k-point convergence criteria. The testing results for k-point are shown below.

The fig.(a) is total energy function with different k-mesh. The total energy converges at 5*15*15 k-mesh. Fig.(b) displays the Seebeck coefficient along a direction with different k-mesh, implying the transport properties converges at 5*15*15 k-mesh. In the manuscript, for HSE06 functional, we employed 5*15*15 k-mesh. We believe the 5*15*15 k-mesh is sufficiently large to reach the convergence.

7. page 6: "Meanwhile, due to $S \propto n^{-3/2}$, the Seebeck coefficient calculated by HSE06 functional is larger than that calculated by GGA-optB88 functional at the same carrier concentration"

I cannot understand this sentence. If the authors would like to use a " $S \propto$

$n^{-3/2}$ " relation, why does the Seebeck coefficient become different "at the same carrier concentration"?

Response:

S is proportional to $n^{-3/2}$, whereas n is decided by chemical potential. The carrier concentration of GGA-optB88 functional is higher than that of HSE06 functional at the same chemical potential, resulting the same carrier concentration of GGA-optB88 and HSE06 functional in fig.4 corresponds to different chemical potentials. We have modified the ambiguous description in the revised manuscript.

8. Fig. 5: Is the energy zero in this figure consistent with that shown in Fig.3? It seems that the energy zeros are different between these figures. Please provide the definition of the energy zero in Fig. 5.

Response:

The energy zeros of fig.3 and fig.5 are different. We modified the energy zero of fig.5 to the top of the valence band, which is consistent with fig.3. The unit of energy in fig.5 should be Ry, not eV. We have modified the two problems in the revised manuscript.

9. The relaxation times presented in Table 1 are partially too optimistic, reaching several hundreds of [fs], and this optimistic value is closely related to the main finding of this manuscript, i.e., the high ZT in the

n-type GeSe. The author should carefully discuss the validity of their theoretical prediction because they use a rather simple approximation for estimating the relaxation time. For example, is it possible to compare their calculation and calculation presented in the study [39] raised by the authors as "consistent" while the details of consistency is not shown in the present manuscript?

Response:

We can't make quantitative comparison between our result and reference [39]. It is a common approximate method to calculate carrier mobility and relaxation time by using deformation potential theory. The electron relaxation time is decided by elastic constant C , deformation potential E_1 based on the deformation potential theory. Although we can't compare our data with other data, we analyzed those parameters in addition to the electron relaxation time in the revised manuscript.

10. The authors estimate the electrical thermal conductivity using the Wiedemann Franz law, but the BoltzTraP code they employed in this study should give the calculated electrical thermal conductivity without using that law. I do not think that there is no need to use the additional approximation (Wiedemann Franz law) here.

Response:

We can only calculate the electrical thermal conductivity within constant

electron relaxation time(ke/τ) with BoltzTraP code. To get the electrical thermal conductivity, the electron relaxation time is also necessary as electrical conductivity. The description of BoltzTraP code shows the calculated electronic thermal conductivity is in good agreement with Wiedemann Franz law [3]. Therefore, Wiedemann Franz law is usually used to calculate the electronic thermal conductivity in many works.

11. Minor issues:

(1) Fig. 1: It seems that the crystal structures are shown using the VESTA software, which requires the appropriate reference while not shown in this manuscript. In the caption of Fig. 1, se -> Se.

(2) Figs. 4, 8: The unit of the Seebeck coefficient [mV/K] seems to be wrong.

Response:

Thanks for proposing these questions. We have responded to them in the revised manuscript.

Reference

- [1]. Yusufu A, Kurosaki K, Kosuga A, Sugahara T, Ohishi Y, Muta H, Yamanaka S. 2011. Thermoelectric properties of Ag_{1-x}GaTe₂ with chalcopyrite structure. *Appl. Phys. Lett.* **99**, 61902. (doi: <http://dx.doi.org/10.1063/1.3617458>)
- [2]. Toberer ES, Zevalkink A, Snyder GJ. 2011. Phonon engineering through crystal chemistry. *Journal of Materials Chemistry* **21**, 15843-15852. (doi: <https://doi.org/10.1039/C1JM11754H>)
- [3]. Madsen GKH, Singh DJ. 2006. BoltzTraP. A code for calculating band-structure dependent quantities. *Comput. Phys. Commun.* **175**, 67-71. (doi: [10.1016/j.cpc.2006.03.007](https://doi.org/10.1016/j.cpc.2006.03.007))

Appendix C

Response to Reviewer 2

Thank you for pointing out the correct analysis method of bonding state and anti-bonding state. This not only improves the reliability of our manuscript, but also be beneficial for us to carry out the related research in the future.

From the PDOS, we analyzed the contribution of states in different energy regions, especially the near the Fermi level, instead of discussing the so-called bonding and anti-bonding properties in the revised manuscript.